# Isolated Variable Domains of an Antibody Can Assemble on Blood Coagulation Factor VIII into a Functional Fv-like Complex

**DOI:** 10.3390/ijms23158134

**Published:** 2022-07-23

**Authors:** Svetlana A. Shestopal, Leonid A. Parunov, Philip Olivares, Haarin Chun, Mikhail V. Ovanesov, John R. Pettersson, Andrey G. Sarafanov

**Affiliations:** Center for Biologics Evaluation and Research, U. S. Food and Drug Administration, 10903 New Hampshire Avenue, Silver Spring, MD 20993, USA; svetlana.shestopal@fda.hhs.gov (S.A.S.); leonid.parunov@fda.hhs.gov (L.A.P.); philip.olivares@fda.hhs.gov (P.O.); haarin.chun@fda.hhs.gov (H.C.); mikhail.ovanesov@fda.hhs.gov (M.V.O.); j@kent.nu (J.R.P.)

**Keywords:** single-chain variable fragment, antibody engineering, coagulation factor VIII, thrombin, low-density lipoprotein receptor-related protein 1

## Abstract

Single-chain variable fragments (scFv) are antigen-recognizing variable fragments of antibodies (FV) where both subunits (V_L_ and V_H_) are connected via an artificial linker. One particular scFv, iKM33, directed against blood coagulation factor VIII (FVIII) was shown to inhibit major FVIII functions and is useful in FVIII research. We aimed to investigate the properties of iKM33 enabled with protease-dependent disintegration. Three variants of iKM33 bearing thrombin cleavage sites within the linker were expressed using a baculovirus system and purified by two-step chromatography. All proteins retained strong binding to FVIII by surface plasmon resonance, and upon thrombin cleavage, dissociated into V_L_ and V_H_ as shown by size-exclusion chromatography. However, in FVIII activity and low-density lipoprotein receptor-related protein 1 binding assays, the thrombin-cleaved iKM33 variants were still inhibitory. In a pull-down assay using an FVIII-affinity sorbent, the isolated V_H_, a mixture of V_L_ and V_H_, and intact iKM33 were carried over via FVIII analyzed by electrophoresis. We concluded that the isolated V_L_ and V_H_ assembled into scFv-like heterodimer on FVIII, and the isolated V_H_ alone also bound FVIII. We discuss the potential use of both protease-cleavable scFvs and isolated Fv subunits retaining high affinity to the antigens in various practical applications such as therapeutics, diagnostics, and research.

## 1. Introduction

A variable fragment (Fv) of an antibody is the smallest structurally distinct antigen-recognizing domain of the molecule. An Fv is composed of the N-terminal variable sub-domains of the heavy and light chains (V_H_ and V_L_) of the antibody interacting weakly in a non-covalent manner. A particular Fv was shown to be unstable at a low concentration indicating the dissociation of V_H_ and V_L_ subunits [1]. To improve the complex stability, the C-terminus of a V_L_ can be genetically connected to the N-terminus of a V_H_ with a linker to make an artificial single-chain Fv (scFv) [2,3]. Several characteristics such as high antigen recognition specificity, a small size (~30 kDa), and a very short plasma half-life (<10 min in mice) make scFvs a unique type of reagent in a broad range of applications including therapeutics, diagnostics, imaging, protein structure research, etc. [4,5,6].

One particular scFv, KM33, was developed against human factor VIII (FVIII), an important protein in the blood coagulation cascade as its functional deficiency results in life-threatening bleeding (hemophilia A). The disease is treated by infusions of therapeutic FVIII, either plasma-derived or recombinant. As a complication, some patients develop antibodies (inhibitors) against FVIII, which render FVIII replacement therapy ineffective [7]. KM33 was derived from an anti-FVIII antibody isolated from an individual with severe hemophilia and was originally expressed in bacteria [8]. In biochemical studies, KM33 was found to bind FVIII with high affinity and inhibit FVIII interactions with its major physiological ligands, and therefore, can be a useful tool in FVIII research.

FVIII is a large heterodimeric protein (~300 kDa) composed of heavy and light chains with the domain structure of A1-A2-B and A3-C1-C2, respectively. FVIII circulates in plasma in a complex with von Willebrand factor (VWF), which stabilizes FVIII by blocking FVIII interactions with its clearance receptors and cell membranes. At sites of blood coagulation (tissue damage), FVIII is activated by a set of site-specific cleavages by thrombin or activated factor X (FXa). Activated FVIII (FVIIIa) dissociates from VWF and acts as a cofactor of activated factor IX. This complex (tenase) is assembled on platelet membranes and activates factor X resulting in eventual blood clotting [9].

Thrombin, a serine protease, is the major activator of FVIII [10] and a key enzyme in the blood coagulation cascade. During activation, thrombin cleaves FVIII after Arg372 (A1-A2 junction), Arg740 (A2-B junction), and Arg1689 (N-terminal region of A3 domain), which results in the formation of an FVIIIa heterotrimer A1/A2/A3-C1-C2 [9]. The initial cleavage site is after Arg740, which subsequently facilitates the cleavage at the other two positions [11], while cleavage after Arg372 is the rate-limiting step of FVIII activation [12,13]. Besides FVIII, thrombin activates other components of the blood coagulation cascade and is involved in fibrinolysis [14,15], inflammation, atherosclerosis [16,17], and neurodegeneration [18,19], where it has numerous protein substrates [20,21].

ScFv KM33 was shown to have an epitope on the FVIII C1 domain [22,23]. The C1 domain is involved in many FVIII interactions including those within the tenase complex, platelets, VWF, and FVIII clearance receptors such as low-density lipoprotein receptor (LDLR) and LDLR-related protein 1 (LRP1) [22,24,25,26,27,28]. When bound to FVIII, KM33 inhibits its activity, cellular uptake, and interactions with VWF, LDLR, and LRP1 [23,24,29,30]. In cell culture, KM33 reduces the immune response against FVIII [29], thus it is used in studies to reduce the immunogenicity of therapeutic FVIII [31]. In mice, KM33 prolongs the plasma half-life of FVIII [32], thus it can be useful for new designs of therapeutic FVIII with extended plasma half-life (EHL) for better treatment of hemophilia A [33]. An insect cell-derived version of the scFv (iKM33) was developed [34] (GenBank JQ797446).

In this study, we aimed to expand the functionality of scFv KM33 in FVIII research. We tested the possibility of reversing inhibition of FVIII functions by bound KM33 upon thrombin cleavage to disintegrate the scFv to explore such a design of therapeutic FVIII. Our results provide new insight into the properties of cleavable scFvs and their fragments and may facilitate the development of new applications involving such molecules.

## 2. Results

### 2.1. Design and Expression of iKM33 Variants with Thrombin Cleavage Site

To generate a thrombin-cleavable iKM33, we introduced a thrombin recognition site into the linker connecting the V_H_ and V_L_ chains of the scFv. We used two variants of such a site: comprised of the Arg740 and Arg372 FVIII sites, and two versions of the first variant with linker lengths of 15 or 17 amino acid residues (Figure 1). Thus, three variants of thrombin-cleavable iKM33 (v1–v3), along with the previously described parental iKM33 [34], were expressed in a baculoviral system and purified using a combination of Ni-affinity and size-exclusion chromatography.

### 2.2. Testing iKM33 Variants for Binding with FVIII

To verify that the structural modifications did not affect the protein’s affinity for FVIII, the generated iKM33 variants were evaluated for FVIII binding. Each protein was immobilized and probed with FVIII in solution using surface plasmon resonance (SPR). All scFv variants including parental iKM33 demonstrated a similar high affinity to FVIII (0.8–1 nM K_D_) (Figure 2), consistent with that found previously [34]. Thus, the structural modifications of iKM33 did not affect its affinity to FVIII.

### 2.3. Testing iKM33 Variants for Thrombin Cleavage

To test the cleavability of the iKM33 variants, they were treated with thrombin, the reaction was stopped by Phe-Pro-Arg-chloromethylketone (PPACK), an irreversible inhibitor of thrombin [35,36], and the samples were analyzed by sodium dodecyl sulfate-polyacrylamide gel electrophoresis (SDS-PAGE). For all cleavable iKM33 variants, the single-chain form (~31 kDa) was reduced into two fragments with expected molecular masses of V_L_ (~17 kDa) and V_H_ (~13 kDa) confirming their cleavage (Figure 3A). Notably, iKM33-v3 was cleaved partially indicating that its cleavage occurred at a slower rate than the two other variants. This agrees with the known fact that FVIII is cleaved by thrombin at Arg372 with a slower rate than that at Arg740 [12,13]. No thrombin cleavage of the parental iKM33 (control) was observed in preliminary experiments.

To test if the cleaved V_H_ and V_L_ subunits dissociate from each other under the used conditions, the samples were analyzed by size-exclusion fast protein liquid chromatography (SE-FPLC) along with uncleaved iKM33-v1 (control) (Figure 3B). The non-treated-by-thrombin sample produced a major peak (~31 kDa) corresponding to the non-cleaved scFv and a minor peak (~150 kDa) corresponding to oligomeric species of protein. In contrast, the thrombin-treated samples produced a single peak eluted from the column at a time corresponding to a mix of the cleaved V_H_ and V_L_ (~13 kDa and ~17 kDa). This demonstrates that upon cleavage of iKM33, its subunits dissociate from each other, which is in agreement with a previous study that used an isolated Fv of a separate antibody [1].

For evaluating the cleavage kinetics of the FVIII/scFv complex, we treated FVIII and iKM33-v2 separately and mixed with different thrombin amounts; the reactions were stopped with PPACK and analyzed by SDS-PAGE (Figure 4A). For thrombin-cleaved FVIII, the pattern of resulting fragments corresponded to the previously described [37,38]. For thrombin-cleaved scFv, fragments corresponded to V_L_ and V_H_, consistent with Figure 3A. To assess the cleavage rates of each protein within their complex, the three cleavage fragments of FVIII (A1, A2, and A3-C1-C2) were tracked due to the presence of three thrombin cleavage sites within FVIII, while only V_L_ of the scFv was tracked considering the presence of one thrombin cleavage site within it (Figure 4B). The results show that the cleavage rate of the sites within FVIII is higher than respective sites within the scFv. This indicates that additional determinants of FVIII contribute to its faster cleavage rates.

### 2.4. Effect of iKM33 Variants on FVIII Activity

Initially, we tested the effects of parental iKM33 on FVIII activity in four different assays: a chromogenic substrate (CS), a thrombin generation (TG), an activated partial thromboplastin time (APTT), and a video microscopy of clot growth assays. In all cases, iKM33 showed strong inhibitory effects on FVIII activity consistent with our previous findings [34], and the strongest effect was observed in the TG assay (Appendix A).

As most sensitive to iKM33 inhibition, the latter assay was then used for evaluation of the effects of the iKM33 variants on FVIII activity. All scFv variants, whether cleavable or not, inhibited FVIII activity in a similar dose-dependent mode (Figure 5). This indicates that upon the cleavage of the scFv variants by thrombin, generated during the reaction, the cleaved fragments remained bound to the cleaved (activated) FVIII.

### 2.5. Effect of Thrombin-Cleaved iKM33-v2 on FVIII Activity

To verify the hypothesis that the cleaved scFv fragments remained bound to FVIII, we tested if a preparation of thrombin-cleaved iKM33 (i.e., a mixture of dissociated V_H_ and V_L_) is capable of inhibiting FVIII activity. In further experiments, we used iKM33-v2 as a representative of all three variants of iKM33. In the first experiment, thrombin-cleaved scFv, confirmed by PAGE, was incubated with FVIII at different molar ratios and the samples were tested for FVIII activity by CS assay. Both thrombin-cleaved and non-cleaved scFvs inhibited FVIII activity dose-dependently, however, achieving the same inhibitory effects by thrombin-cleaved scFv required using their approximately 100-times higher molar excess over FVIII (Figure 6A). To verify that PPACK, present in the reaction to inactivate thrombin, does not affect the CS assay, we also tested a sample of FVIII incubated with PPACK-treated thrombin that did not inhibit the reaction. This confirmed that PPACK did not affect FVIII activity at the used concentration.

In the second experiment, the iKM33-v2 thrombin cleavage was stopped with biotinylated PPACK. The inactivated thrombin, as well as excess PPACK, were removed using streptavidin-coated beads. The preparation was confirmed for scFv cleavage by SDS-PAGE after incubation with FVIII at different molar ratios and tested for FVIII activity by TG assay. Like the previous experiment, the preparation of thrombin-cleaved scFv inhibited FVIII activity in a dose-dependent manner (Figure 6B). Compared to the first experiment, the effect of thrombin-cleaved scFv was significantly more pronounced, achieving the same levels of inhibition at approximately 5–6 times lower molar excess over FVIII. Such a difference can be attributed to the approximately 6–7 times higher protein concentrations in the TG assay and other conditions. Altogether, the results indicate that isolated V_H_ and V_L_ subunits bound to FVIII reproduce the inhibitory effect of intact iKM33.

### 2.6. Effect of Thrombin-Cleaved iKM33-v2 on Binding FVIII to LRP1

To verify the hypothesis that the V_H_ and V_L_ can assemble on FVIII, we tested the effect of the subunits on the binding of FVIII to LRP1. Previously, KM33 was shown to inhibit LRP1-mediated internalization of FVIII in cell culture, indicating blockage of this interaction [29]. Thus, LRP1 was immobilized and tested for interactions with FVIII in the presence of increasing concentrations of thrombin-cleaved and non-cleaved iKM33-v2 by SPR. Both preparations inhibited FVIII binding to LRP1 (Figure 7) consistent with the previous cell culture results and supporting the assembly of the V_H_ and V_L_ on FVIII to mimic the binding of the intact iKM33. Notably, the inhibitory effects in this assay were significantly more pronounced than in FVIII activity assays, most likely due to significantly higher concentrations of the ligands (by 10^3^–10^4^ times). The results also show that a site of FVIII involving its epitope for KM33 [22] and/or a site(s) proximal to it is important for the interaction with LRP1.

### 2.7. Testing Interaction between FVIII and Isolated iKM33 Subunits

Finally, we tested whether the isolated V_H_ and V_L_ alone or together can interact with FVIII. We took advantage of the 10 × His tag in the V_L_ domain (Figure 1) that allowed separation of the subunits from thrombin-cleaved scFv preparations using a Ni-affinity sorbent. Then, a recombinant B-domain deleted FVIII (BDD) was incubated with each of the isolated or combined V_H_ and V_L_, or with thrombin-cleaved or non-cleaved scFv (controls) in a pull-down assay using an FVIII-affinity resin. In a dose-dependent fashion, FVIII-containing samples carried over the intact and thrombin-cleaved scFv, a mix of isolated V_H_ and V_L_, and isolated V_H_, but not V_L_, analyzed by SDS-PAGE (Figure 8). In samples not containing FVIII, no scFv and related proteins were carried over showing that their transfer occurred via FVIII, but not the resin matrix (Figure 8). This shows that when taken together, both subunits (V_H_ and V_L_) bound FVIII, as well as the V_H_ subunit taken alone, whereas the V_L_ taken alone did not bind FVIII. In turn, the binding of isolated V_H_ to FVIII promoted the binding of isolated V_L_ to the FVIII/V_H_ complex, indicating that both subunits interact and form an scFv-like complex on the FVIII molecule. Considering all our data, this, in turn, indicates that the V_H_/V_L_ complex interacts with the same FVIII epitope as that for the intact scFv.

## 3. Discussion

In the present study, we investigated the properties of iKM33 containing a thrombin cleavage site in the linker connecting the V_H_ and V_L_ subunits. Initially, we tested whether this scFv dissociates from the FVIII complex upon thrombin cleavage. This was assessed by the analysis of proteolytic fragments of the thrombin-treated complex by SDS-PAGE and measurement of FVIII activity in the sample. Unexpectedly, we found that the cleavage of the scFv did not alter its inhibitory effect on FVIII activity and proposed that the cleaved scFv fragments did not dissociate from FVIII. To verify this hypothesis, we tested the effects of the thrombin-cleaved scFv in an FVIII activity assay and FVIII binding assay with LRP1 and found that the scFv preparation inhibited both FVIII properties. Using preparations of V_H_ and V_L_ subunits separated from each other, we demonstrated their ability to bind FVIII when used altogether. We concluded that both subunits reassembled into the scFv-like heterodimer on the same epitope on FVIII as for the intact scFv. Speaking more generally, the antigen catalyzed an assembly of the antibody’s isolated variable domains into a functional Fv-like complex. Notably, antigen-dependent stabilization of the complex of antibody variable fragments was previously used for the development of open-sandwich immunoassays for the detection of some ligands [39,40]. We also found that the isolated V_H_ alone bound FVIII with an affinity comparable with that of the scFv.

An important facet of the study design was the ability of thrombin-cleaved iKM33 to disintegrate into isolated V_H_ and V_L_ subunits evidenced by shifting the protein elution time compared to the intact scFv by SE-FPLC. These data are consistent with a previous study that found K_D_ in a micromolar range for V_L_ and V_H_ of a different Fv [1], and that this may reflect a property of some antibodies to have relatively low affinity between the V_L_ and V_H_ subunits. In testing the effect of thrombin-cleaved iKM33 on FVIII binding to LRP1, we found that the binding inhibition was similar to that of the intact iKM33. In contrast, inhibition of FVIII activity by the thrombin-cleaved iKM33 was less pronounced compared to uncleaved iKM33. Most likely, this was caused by significantly lower ligand concentration in the activity assay (by 10^3^–10^4^ times) that favored dissociation of the subunits from FVIII. Between both activity assays used (CS and TG), we also observed variations in the inhibitory effect due to differences in conditions. However, all results supported the assembly of the isolated subunits on the iKM33 epitope on FVIII. Notably, the V_H_ that showed a higher affinity to FVIII, was derived from an anti-FVIII antibody of native origin, whereas the V_L_ was derived in vitro using a gene library of non-immune origin [8].

We want to emphasize that all iKM33 modifications made did not affect its major properties based on testing scFv variants in FVIII binding and activity assays. At the same time, the variants of iKM33 showed differences in thrombin cleavage rates as both variants bearing a thrombin cleavage site identical to that with Arg 740 in FVIII were cleaved faster than the variant bearing the site identical to that with Arg 342 in FVIII. This observation is in accordance with the same difference in cleavage rates of the respective sites within FVIII [12,13]. At the same time, thrombin cleavage of FVIII being much faster than of all iKM33 variants strongly suggests that additional determinants in FVIII are responsible for this difference. Most likely, these are sulfated tyrosine residues adjacent to the FVIII thrombin cleavage sites previously suggested to affect the cleavage rates [41].

Our results suggest that the use of protease-cleavable scFv has potential for the development of new in vivo and in vitro applications. The in vivo usage may relate to acquiring activity or increasing the efficacy of some therapeutic proteins upon the cleavage [42,43]. Examples include protease-activatable drugs (pro-drugs), antibodies (pro-antibodies), probes for imaging, and conjugates for photodynamic therapy. Notably, thrombin-cleavable drugs are under development to target the activity of the drug at sites where thrombin presents, such as the sites of clot formation, injury, and inflammation [44,45,46,47]. An example of a thrombin-activatable pharmaceutical closely related to our study (currently in clinical trials) is an EHL FVIII representing a recombinant FVIII genetically fused to the FVIII-binding domain of VWF [48]. Though such use of thrombin-cleavable iKM33 is unlikely due to the extremely high affinity to FVIII resulting in the retention of the cleaved scFv subunits on FVIII, the lesson learned from our study indicates that the affinity of the fused protein-ligand to FVIII should be moderate to allow dissociation of the parts upon the cleavage of the chimeric molecule. For a high-affinity scFv, a possible approach to reduce the affinity could be using point mutations including those within the V_H_/V_L_ interface to destabilize the subunits’ interaction.

The in vitro usage of protease-cleavable scFvs may be based on the ability of separated V_H_ and V_L_ to assemble on the target antigen and allow its detection with low background signal. Such an approach can utilize Förster Resonance Energy Transfer (FRET), which is based on distance-dependent energy transfer between a donor dye and relevant acceptor dye, typically in the range of 1–10 nm [49,50]. In this application, each subunit of the scFv can be labeled with the respective dye, so both would make a specific FRET dye pair. Increased energy transfer would be observed when both scFv subunits associate with the antigen and the resulting acceptor dye emission signal would correlate with the antigen level. Similarly, a quench-based system can be designed where the fluorescence is quenched as a quencher-labeled subunit binds near the dye-labeled subunit. The energy transfer to the quencher would result in non-radiative decay and loss of signal rather than a change in the emission spectrum. A less complex alternative would be to apply Protein-Induced Fluorescence Enhancement (PIFE). PIFE can be applied over short distances within the 0–3 nm range as the fluorescence of some dyes increases up to four-fold when in close proximity to a protein [51]. To apply such an approach, one of the subunits can be labeled in proximity to other proteins upon the formation of the antigen/V_L_/V_H_ complex. Lastly, the isolated V_L_ and V_H_ fragments interacting with respective antigens can be potentially used for the development of lateral flow immunoassays for rapid testing in medicine, agriculture, food, and environmental sciences [52,53].

Furthermore, the ability of an isolated scFv subunit to bind the antigen (such as V_H_ of iKM33 in our study) in the absence of another subunit indicates additional usability of this type of molecule. Indeed, the variable subunit is the smallest fragment (MW of ~13 kDa) of antibody retaining the ability to recognize the antigen, in particular, two times smaller than the scFv (MW of ~30 kDa). Such a subunit is structurally homologous to the variable domains of the heavy chain-only antibodies (V_HH_) found in camelids termed nanobodies. These small (of nanometric size) molecules were proven to have good stability, solubility, and tissue permeability which makes them superior reagents in various applications including cell biology, diagnostics, and therapeutic use [54,55,56,57,58]. We believe that the similar use of isolated variable subunits from common antibodies (V_H_ or V_L_), retaining high affinity to the antigen, in particular, in biochemical competition assays and imaging may result in higher mapping precision for ligand binding sites and image quality with faster sample processing. However, the use of isolated V_H_ or V_L_ subunits may require additional modifications of the molecules to improve their physicochemical characteristics [59]. Future studies will investigate the practical applicability of both thrombin-cleavable scFvs and the single variable subunits.

## 4. Materials and Methods

### 4.1. Reagents

Recombinant FVIII and BDD FVIII were purchased as FVIII pharmaceutical products from ASD Healthcare, AmerisourceBergen (Conshohocken, PA, USA). LRP1 was kindly provided by Dr. D. Strickland (University of Maryland, Baltimore, MD, USA). Other significant reagents were congenital FVIII-deficient plasma (HRF, Inc., Durham, NC, USA); Tris-BSA buffer (HYPHEN BioMed, Neuville-sur-Oise, France); human alpha thrombin, Phe-Pro-Arg-chloromethylketone (PPACK), biotinylated PPACK, corn trypsin inhibitor, activated coagulation factor FXI (Haematologic Technologies, Essex Junction, VT, USA); phospholipid vesicles (Rossix, Mölndal, Sweden); fluorogenic thrombin substrate Z-Gly-Gly-Arg-AMC (Bachem, Bubendorf, Switzerland); Tissue factor Recombiplastin (Instrumentation Laboratory, Bedford, MA, USA), Dynabeads M-280 Streptavidin (Thermo Fisher Scientific, Waltham, MA, USA); and thrombin activity calibrator (Diagnostica Stago, Parsippany, NJ, USA).

### 4.2. Generation of Vectors for Protein Expression

The expression cassettes for iKM33-v1, iKM33-v2, and iKM33-v3 variants were generated based on the iKM33 gene sequence (GenBank JQ797446) [34] where the sequences of linkers between V_H_ and V_L_ subunits were modified as shown on Figure 1. All coding sequences were optimized for *Spodoptera frugiperda* using the GeneOptimizer algorithm (Thermo Fisher Scientific) and synthesized and cloned into pFastBacT1 vector (Thermo Fisher Scientific) using the service of GenScript (Piscataway, NJ, USA). The coding sequences were deposited to GenBank (MW700080, MW700081, and MW700082).

### 4.3. Protein Expression and Purification

Recombinant baculoviruses were generated using the Bac-to-Bac Baculovirus Expression System (Thermo Fisher Scientific). Proteins were expressed from the virus stocks using Sf9 cells cultivated at 27 °C in Sf-900 II SFM media in shake flasks, where the optimal ratio between the number of cells and the volume of viral stock was preliminarily determined on a small scale [60]. Conditioned media from the cultures was collected 72 h after infection. Proteins were purified using an affinity-based step using HisTrap Excel column, followed by a gel-filtration (SE-FPLC) step using Superdex 75 10/300 column (Cytiva, Marlborough, MA, USA), as was described previously [61]. The protein concentration was determined by the absorbance at 280 nm.

### 4.4. Cleavage of scFv Variants by Thrombin and Isolation of Individual V_H_ and V_L_ Subunits

FVIII and scFv iKM33 variants were incubated with thrombin in 10 mM HEPES pH 7.4, and 150 mM NaCl at the conditions specified in individual experiments. The reactions were terminated by the addition of PPACK at 5-fold molar excess over thrombin. The cleavage was confirmed by SDS-PAGE gel analysis in every experiment. For the generation of a thrombin-cleaved scFv preparation for use as an inhibitor of FVIII activity in the TG assay, the scFv was cleaved by thrombin, the reaction was terminated by adding biotinylated PPACK, followed by incubation with magnetic Dynabeads M-280 Streptavidin for 30 min, and removal of beads with a magnet resulting in the removal of both free and thrombin-bound PPACK.

For the preparation of isolated scFv subunits, 36 µg of cleaved iKM33 was added to a microtube with 10 µL of Ni-NTA agarose (Qiagen, Germantown, MD, USA). Upon 30 min incubation on a rotator, the supernatant with a separated V_H_ subunit was collected. The remaining agarose beads were washed with 50 mM Tris-HCl, 300 mM NaCl, 10 mM imidazole, pH 8.0, and protein (V_L_ subunit) was eluted with the same buffer modified to have 250 mM imidazole. The preparations of both V_H_ and V_L_ subunits had buffer exchanged to 10 mM HEPES, 150 mM NaCl, 5 mM CaCl_2_, 0.005% Polysorbate-20, pH 7.4 (HBS-P/Ca) buffer and were concentrated with Amicon Ultra 3 K 0.5 mL centrifugal filters (MilliporeSigma, Burlington, MA, USA). The purity of protein preparations was confirmed by SDS-PAGE analysis.

### 4.5. Sodium Dodecyl Sulfate-Polyacrylamide Gel Electrophoresis

Samples were analyzed on NuPAGE 4–12% Bis-Tris protein gels (Thermo Fisher Scientific). The gels were stained either using the SilverQuest Silver Staining Kit or with SimplyBlue SafeStain (Thermo Fisher Scientific). The densitometry analysis of gels was performed using Image Studio Lite software (LI-COR Biosciences, Lincoln, NE, USA).

### 4.6. Size-Exclusion Chromatography Assay

The scFv preparations were analyzed by Superdex 75 5/150 GL column with Äkta purifier (Cytiva). The column was loaded with 25 µg of protein and chromatography was conducted using HBS-P/Ca buffer at a flow rate of 0.3 mL/min. Fractions were monitored using UV absorption at 280 nm. The molecular weight of protein species in the fractions was assessed using a chromatography standard 151-1901 (BioRad, Hercules, CA, USA).

### 4.7. Surface Plasmon Resonance Assay

The experiments were performed using a Biacore T200 instrument (Cytiva). Proteins were covalently immobilized on CM5 sensor chips in 10 mM sodium acetate pH 4.0 to the level of 250–350 RU using amine coupling. A control flow path cell was blocked with 1 M ethanolamine (blank surface). FVIII was concentrated and buffer exchanged to 10 mM HEPES, 300 mM NaCl, 5 mM CaCl_2_, 0.005% Polysorbate-20 with Amicon Ultra-0.5 100 K centrifugal units (MilliporeSigma) followed by dilution with the same buffer but without NaCl to adjust its final concentration to 150 mM. Binding experiments were performed at a flow rate of 20 µL/min at 25 °C for 3 min of association phase followed by replacing the solution with buffer only for dissociation phase, and the chips were regenerated with 0.1 M phosphoric acid before further injections. The binding signals were analyzed with Biacore T200 evaluation software version 3.2: the signals were double subtracted with the signals from the blank surface and buffer only, and the K_D_ values were estimated using the Langmuir (1:1) binding model.

### 4.8. Factor VIII Activity Assays

APTT clotting assays and video microscopy of clot growth assays were performed as described [34]. CS assays were performed with Coatest SP4 Factor VIII kit (Chromogenix; Diapharma, West Chester, OH, USA). An in-house FVIII standard was used as a calibrator. All samples were diluted in FVIII-deficient plasma to ~1 IU/mL, and then serially diluted in a buffer included in the kit.

For the TG assay, citrated FVIII-deficient plasma (50% vol/vol) was mixed with corn trypsin inhibitor (50 μg/mL), fluorogenic substrate Z-Gly-Gly-Arg-AMC (800 μM), phospholipid vesicles (4 μM) and tissue factor (0.2 pM). Plasma was warmed to 37 °C, and TG in plasma was activated by the solution of CaCl_2_ and activated coagulation factor XI (10 mM and 10 pM final concentrations, respectively). Immediately after the activation, plasma was mixed with the samples (35% vol/vol) containing FVIII and scFv iKM33 variant or thrombin calibrator. AMC fluorescence (460 nm) was recorded at 37 °C with a microplate reader. Thrombin peak height was used to characterize TG as described [62].

### 4.9. Pull-Down Assay with FVIII and iKM33 Fragments

FVIII (BDD) and iKM33 or its fragments at specified amounts were incubated in HBS-P/Ca buffer for 1 h at room temperature in microcentrifuge tubes, incubated with VIIISelect resin (Cytiva), and centrifuged at 12,000× *g* for 1 min. Supernatants were removed, the resin was washed two times with HBS-P/Ca buffer, and bound proteins were eluted with 20 mM HEPES, 900 mM Arginine-HCl, 5 mM CaCl_2_, 45%, propylene glycol, 0.005% Polysorbate-20, pH 6.5 and analyzed by SDS-PAGE with silver staining.

## 5. Conclusions

We developed an scFv (iKM33) directed against FVIII with a thrombin-cleavable linker connecting variable subunits V_H_ and V_L_ and confirmed their dissociation upon cleavage. In the presence of FVIII, both isolated subunits assembled into a functional scFv-like complex, corresponding to the parental Fv fragment and interacting with the same FVIII epitope. The data suggest that the ability of isolated V_H_ and V_L_ subunits to reassemble on respective antigens is a common property of antibodies, more pronounced for high-affinity ligand interactions. Our results may be useful for the development of applications based on protease-cleavable proteins such as drugs, diagnostics, and analytical tools.

## Figures and Tables

**Figure 1 ijms-23-08134-f001:**
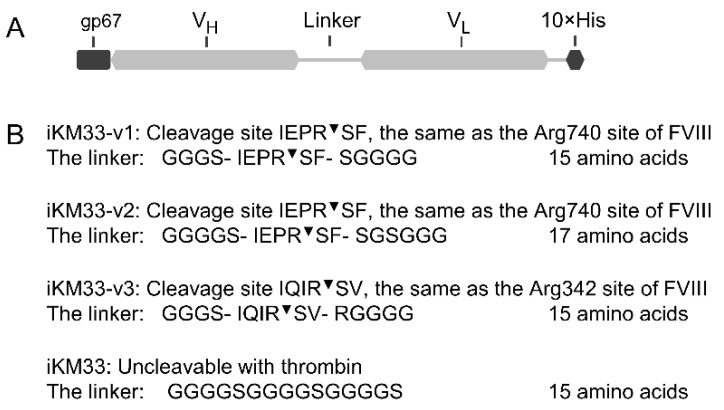
Design of iKM33 variants cleavable by thrombin. (**A**) Structural diagram of the constructs, which include a signal peptide gp67, V_H_ and V_L_ domains connected by a linker with thrombin cleavage site, and 10 × His tag for purification. The signal peptide is cleaved off during protein secretion upon expression in Sf9 cells. (**B**) Amino acid sequences of the linkers in three cleavable scFv variants and parental (uncleavable) iKM33 [34]. The cleavage positions are indicated (^▼^).

**Figure 2 ijms-23-08134-f002:**
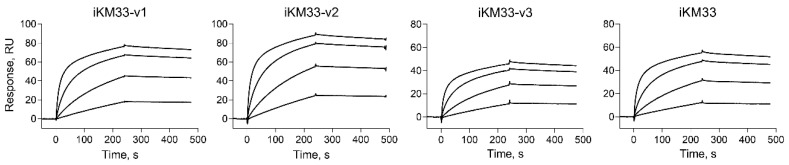
Binding of FVIII to iKM33 variants by SPR. Each scFv variant was covalently immobilized on a sensor chip. FVIII was injected over the chip at concentrations of 3.125 nM, 12.5 nM, 50 nM, and 200 nM. Resulting K_D_ values were calculated using the 1:1 binding model.

**Figure 3 ijms-23-08134-f003:**
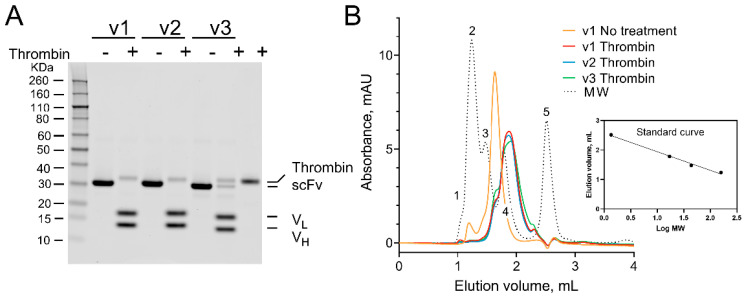
Cleavage of iKM33 variants by thrombin. The scFv variants were incubated with thrombin for 10 min at 37 °C. The samples of iKM33-v1 (v1), iKM33-v2 (v2), and iKM33-v3 (v3) were treated with 0.05 µg of thrombin per 1 µg of protein substrate. The reactions were stopped by adding PPACK. The parental iKM33 (31 kDa) was found not to be cleavable in preliminary experiments. (**A**) The samples with (+) and without thrombin cleavage (−) were loaded on SDS-PAGE gels (3 µg protein per well), and the gel was stained by Coomassie blue. The bands of V_H_ (~13 kDa) and V_L_ (~17 kDa) were identified based on their theoretical molecular weight. (**B**) SE-FPLC analysis of the thrombin-cleaved samples. MW—elution profile of molecular weight markers: thyroglobulin bovine 670 kDa (1), γ-globulin bovine 158 kDa (2), ovalbumin chicken 44 kDa (3), myoglobin horse 17 kDa (4), and vitamin B_12_ 1.35 kDa (5). The insert shows a standard curve based on the markers 2–5.

**Figure 4 ijms-23-08134-f004:**
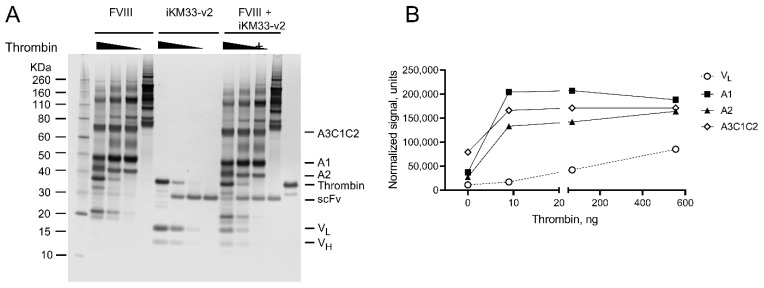
Thrombin cleavage of FVIII and iKM33-v2. (**A**) Samples with FVIII (0.5 µg per sample), iKM33-v2 (0.055 µg per sample), and a mix of FVIII (0.5 µg per sample) and iKM33-v2 (0.055 µg per sample) at a molar ratio of 1:1 were incubated for 10 min with increasing amounts of thrombin (0, 8.7, 69, and 555 ng per sample). The reactions were stopped by PPACK, and the samples were analyzed by SDS-PAGE followed by gel silver staining. The identities of the selected bands are indicated as follows: (i) FVIII fragments: A1, A2, A3-C1-C2, and (ii) iKM33-v2 fragments: scFv—single-chain form, V_H_ and V_L_ subunits. (**B**) Quantitative densitometry of selected FVIII and iKM33-v2 fragments (representative results from one of three experiments).

**Figure 5 ijms-23-08134-f005:**
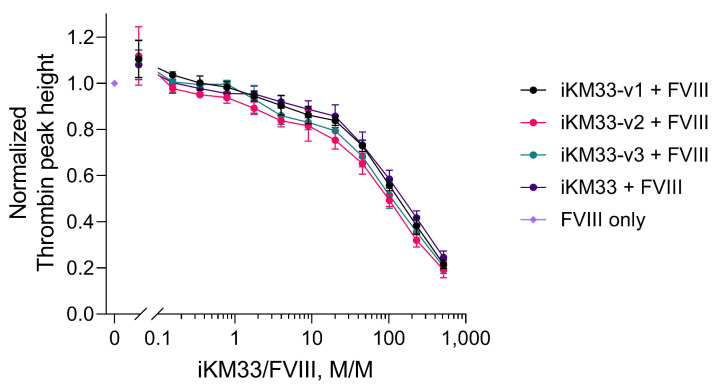
Effects of scFv iKM33 variants on FVIII activity in TG assay. Serially diluted iKM33 variants (60 nM to 0.008 nM) were incubated for 15 min at room temperature with 116 pM FVIII and tested in TG assay. The results are normalized to the control sample (no iKM33). Mean ± SD, *n* = 3. Statistical analysis: two-way ANOVA, Tukey’s multiple comparisons test for each dilution, no significant difference, *p* > 0.05.

**Figure 6 ijms-23-08134-f006:**
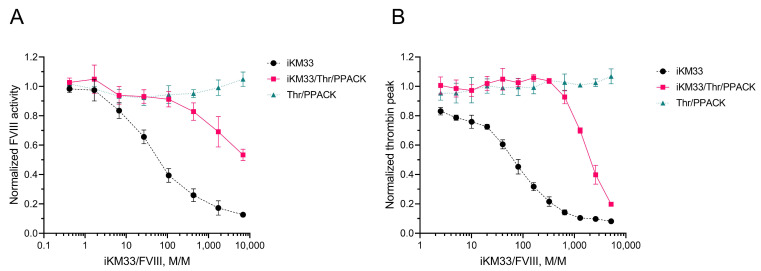
Effect of thrombin-cleaved iKM33-v2 on FVIII activity. (**A**) Three samples were prepared as (i) scFv, (ii) scFv treated with thrombin followed by termination of reaction by PPACK (scFv/Thromb/PPACK), and (iii) buffer treated with thrombin followed by addition of PPACK (control) (Thrombin/PPACK). Serially diluted samples (iKM33 concentration from 1000 to 0.24 nM) were incubated with FVIII (145 pM) and tested by CS assay. The results were normalized to FVIII incubated with thrombin-treated buffer. Mean ± SD, *n* = 4. (**B**) The samples were prepared similarly to panel A, but thrombin was removed from the reactions using biotinylated PPACK followed by incubation with streptavidin beads. Serially diluted samples (iKM33 concentration from 600 to 0.3 nM) were incubated with FVIII (116 pM) for 15 min and tested by TG assay. The results were normalized to FVIII incubated with Tris-BSA buffer (pH 7.2). Mean ± SD, *n* = 3.

**Figure 7 ijms-23-08134-f007:**
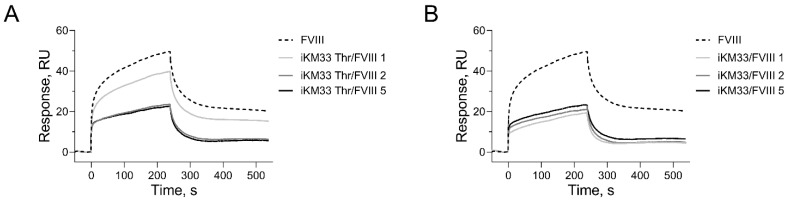
Effect of thrombin-cleaved iKM33-v2 on the binding of FVIII to LRP1. In SPR assay, LRP1 was immobilized on a chip via amine coupling and tested for binding with FVIII (100 nM) preincubated without or with iKM33-v2 at a ratio of 1, 2, or 5 (**A**). In a similar set-up, immobilized LRP1 was tested with thrombin-cleaved iKM33-v2 (reaction was terminated by PPACK and confirmed for cleavage by SDS-PAGE) (**B**). Both experiments were repeated three times with similar results.

**Figure 8 ijms-23-08134-f008:**
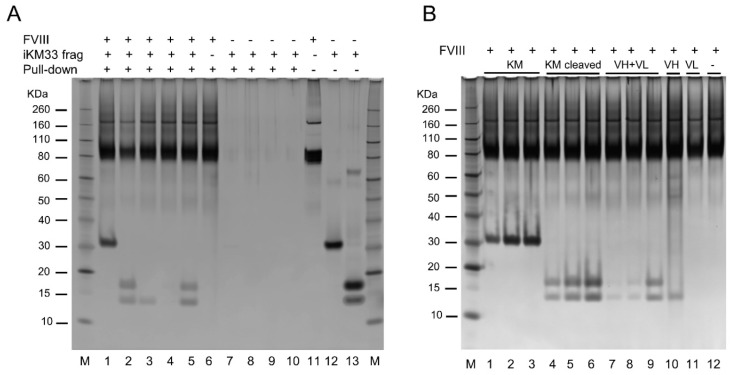
Testing interactions between FVIII and isolated iKM33 subunits. The interaction was evaluated by a pull-down assay using FVIII (BDD), iKM33-v1, and its thrombin-cleaved fragments (as shown on top) and FVIII-affinity sorbent; the resulting samples were analyzed by SDS-PAGE followed by gel silver staining. (**A**) Lanes 1–10 contain the samples, and lanes 11–13 contain control proteins. Specifically, lanes 1–6 contain FVIII (~2 µg), lanes 1–5 and 7–9 contain the scFv and its fragments (used at 3-fold molar excess over FVIII in pull-down assay) as follows: lanes 1 and 7 contain the scFv; lane 2 contains the thrombin-cleaved scFv, lanes 3 and 8 contain the V_H_, lanes 4 and 9 contain the V_L_, and lanes 5 and 10 contain a mixture of the V_H_ and V_L_. (**B**) Analysis of dose-dependence of the scFv fragments used in a similar experiment: lanes 1–12 contain FVIII (~2 µg), lanes 1–9 contain the scFv, thrombin-cleaved scFv, and a mixture of V_H_ and V_L_. In each group shown on top, the molar ratio of FVIII to the scFv fragments increases as 1:1, 1:2, and 1:5. Lane 10 contains the V_H_ and lane 11 contains the V_L_; both subunits were used at 5-fold molar excess over FVIII. In both panels, lane M contains a molecular weight marker. Each experiment was performed independently two times with similar results.

## Data Availability

The data presented in this study are available in the article.

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
