# Peer review of "Isolated Variable Domains of an Antibody Can Assemble on Blood Coagulation Factor VIII into a Functional Fv-like Complex"

_ijms, 2022, doi:10.3390/ijms23158134_

Round 1

Reviewer 1 Report

The revised version provides improvements of data presentation and includes comments on number of experimental replicates performed, which allows the readers to build a critical opinion on the presented data set. Importantly, certain previously vague points of inference have been improved by rephrasing the text. Discussion section has been implemented with clearer interpretations of the observations and more comments on future prospects of possible utility of the novel concept. With that, I can recommend the manuscript for publication.

Author Response

Our Response

We are thankful to reviewer for provided critique. It allowed us to improve the manuscript, and we believe that now it is acceptable for publication.

Reviewer 2 Report

>Furthermore, these data indicate that the variable fragments of other antibodies may behave similarly in presence of respective antigens. To the best of our knowledge, such a phenomenon has not been described in the literature so far. 

I think this is incorrect. Antigen-mediated Fv formation is well known. Based on this phenomenon, Fv screening method was developed, called the open-sandwich method (reference like, Anal. Chem. 2009, 81, 20, 8298–8304 "Open-Sandwich Enzyme Immunoassay for One-Step Noncompetitive Detection of Corticosteroid 11-Deoxycortisol" ). 

>Furthermore, the ability of only one isolated subunit to bind the antigen in absence of another subunit of the scFv (as found for VH of iKM33 and FVIII) indicates that this could be representative of a new class of ligands which are significantly smaller than the typical scFv

This is also well known as VH antibody and nothing novel (reference like,Methods Mol Biol. 2009;525:187-216 "Selection of non-aggregating VH binders from synthetic VH phage-display libraries" ). Camel also has a small variable domain called VHH, which was widely investigated for several application purposes.

Thus, the authors must investigate the previous studies and revise the manuscript to include the above well know issues in the manuscript. 

Author Response

>Furthermore, these data indicate that the variable fragments of other antibodies may behave similarly in presence of respective antigens. To the best of our knowledge, such a phenomenon has not been described in the literature so far. 

I think this is incorrect. Antigen-mediated Fv formation is well known. Based on this phenomenon, Fv screening method was developed, called the open-sandwich method (reference like, Anal. Chem. 2009, 81, 20, 8298–8304 "Open-Sandwich Enzyme Immunoassay for One-Step Noncompetitive Detection of Corticosteroid 11-Deoxycortisol"). 

Our Response

  1. We agree and acknowledge that we missed the literature describing the antigen-stabilized variable domain complex formation and use of this principle in such application as open-sandwich ELISA for detection of ligands. In the revised paper, we provided description of this use with three respective references. Please find this description in the first paragraph of the revised Discussion (marked version, in red font).
  2. In both the initial and revised papers, we are not claiming that such a phenomenon (in general) was first discovered by us, but rather describe that, in regard to the specific antigen-scFv pair (FVIII and iKM33, respectively). We believe that our results will be useful for (i) knowledge of general structure-function relationship of an antibody-antigen, (ii) future designs of protease (thrombin etc.)-cleavable therapeutics (as discussed in paper) and (iii) FVIII field.
  3. Regarding dissociation of the variable subunits of an Fv (scFv) upon its disintegration, we believe that such a phenomenon has not been described in the literature (at least clearly). The only one paper we found stated that a particular Fv was “unstable at a low protein concentration” indicating its dissociation into the VH and VL subunits (ref. #1 in paper). Thus, we believe that the observation we made is useful in both general and FVIII/KM33 fields.

>Furthermore, the ability of only one isolated subunit to bind the antigen in absence of another subunit of the scFv (as found for VH of iKM33 and FVIII) indicates that this could be representative of a new class of ligands which are significantly smaller than the typical scFv.

This is also well known as VH antibody and nothing novel (reference like, Methods Mol Biol. 2009;525:187-216 "Selection of non-aggregating VH binders from synthetic VH phage-display libraries"). Camel also has a small variable domain called VHH, which was widely investigated for several application purposes.

Thus, the authors must investigate the previous studies and revise the manuscript to include the above well know issues in the manuscript. 

Our Response

We agree that the binding of isolated VH subunits to the antigen is known, as in particular was tested in the referenced study. In both our initial and revised papers, we did not claim that we discovered that for the first time, but rather describe this phenomenon regarding to FVIII and KM33.

Per your comment, in the revised paper, we provide an overview of practical applicability of the nanobodies (VHH) and suggest that VH subunits from common antibodies, as structure-function homologs may have similar practical use. Please find this description in the last paragraph of the revised Discussion (marked version, in red font).

In summary, we are thankful to the reviewer for all these comments. We believe that we have adequately addressed them in the revised manuscript that improved its quality, and now the manuscript is acceptable.

This manuscript is a resubmission of an earlier submission. The following is a list of the peer review reports and author responses from that submission.

Round 1

Reviewer 1 Report

In this manuscript, Shestopal et al. demonstrated evaluations of antibody fragments against Factor VIII. The authors used scFv format of anti FVIII antibody iKM33 and introduced thrombin cleavage sites in the middle of the linker with several variations (iKM33-v1~v3). These scFvs retained the FVIII binding activity confirmed by Biacore analysis. Thrombin digestion cleaves the linker and the resulting VH and VL existed as monomers confirmed by size exclusion chromatography. Binding inhibition assay and pulldown assay revealed that the cleaved scFv, i.e. VH and VL, showed binding activity toward FVIII. The authors speculated that the VH and VL of iKM33 formed heterodimer through the binding toward FVIII because size exclusion chromatography showed that cleaved scFv could not form Fv dimer.

This reviewer speculates that the initial aim of this study may be the binding control of iKM33 scFv upon thrombin digestion. Unfortunately, the cleavage of the linker did not alter anything and cleaved scFv still has the binding activity as same as its scFv format. Thus, it seems that the results showing here is nothing novel. These thrombin cleavable scFvs have no-use for any applications. Thus, this reviewer cannot support its publication in the IJMS.

Reviewer 2 Report

Minor points:

Line

11 (scFv) are variable

33 shown to be unable

72 reduce the immune

86 FVIII, and two

116 than the two

128 Fv of

225 a site proximal

235 advantage of trhat the

238 In adose-

246 on the FVIII

246 all our

259 upon using isolated VH

272 dependence

281 that the affinity

303 they correlated

323 in the range

333 such an approach

376 magnetic Dynabeads

406 at a low

418 in the

425 with a mocroplate

437 in the presence

439 that the ability

441 results may be useful

Reviewer 3 Report

In the present manuscript, the authors present the activity of engineered scFv with inhibitory activity against Factor VIII. The thrombin cleavage sites introduced into the scFv enabled the dissociation of VL and VH subunits. Interestingly, thrombin-cleaved subunits were still inhibitory, and so it was concluded that the isolated subunits are able to assemble in the presence of the antigen. The conceptualization of the study is very thorough, experiments are carefully designed and performed and results clearly presented. Also experimental details are well described. The reasoning behind the experiments is clearly outlined. My only major comment would be that the strength of VH/VL interaction of individual scFvs should not be generalized, because it is well documented that they interact with different strength in different fragments. A valuable point for discussion would be if the formation of ternaty complex on FVIII is still expected if the contact between VH and VL is weakened (e.g. using destabilizing point mutations)? On another note, is there anything known on the interaction of the studied scFv with FVIII – are there data (from e.g. mutagenesis or structural studies) supporting the observed strong VH only-antigen interaction in the pull-down assay?

Please find below a list of minor remarks which I hope you will find helpful.

Line 129, Figure 3B. molecular weight markers (MW standard) to monitor the SEC run should be shown

Figure 4a and 4b: is it possible to show the results of multiple independent experiments and introduce the deviation bars?

Line 232 (Figure 7B): The Figure Legend is difficult to understand: are these averages of the experiment, presented at the beginning oft he Figure Legend? Is only the v2 variant presented?

Line 236: BDD (B-domain deleted) abbreviation explanation does not occur until Material and Methods section

Line 242: This sentence is not clear: „both subunits indeed bound“ – I think this should be reworded to „VH and VL subunits linked“, or similar

Line 247: Figure 8 should be moved to the results section

Line 257: „upon the using isolated the VH and VL subunits“, this sentence is not clear, please reword

Line 261: maybe reword: that antigen binding alone can incite the binding of antibody‘s single variable domains to elicit a function similar as exhibited by an integral Fv-like fragment.

Lines 301-304: „there is one sulphated tyrosine by…“ – do you mean in the proximity? The next case in the line 302. „and the correlated with a faster cleavage“ – the sentence not clear, please correct.

Line 312: citation of the clinical study would be appropriate

Line 316: „remnants“ – would components maybe be a better description?

Line 317: The reasoning here is difficult to follow. Do you mean that other Factor VIII-inhibitory scFvs, where there is less affinity between VH/VL, should be used for these purpose? Or should the inhibitory ligand dissociate when Factor VIII is cleaved?

Line 335: do you mean cleavable scFv fragments? Do the described assays not rely on very stable reagents, and is there any data on storage stability of protease-cleaved subunits of scFvs?

Line 366: It would be more appropriate to describe the purification steps as: affinity based IMAC chromatography using His-Excel column, and gel filtration using Superdex75.

Line 384: HBS-P buffer is defined first in lines 395-396

Line 418: in the kit

Figure S1: please amend the legend by including the explanation of the abbreviations. D: um should be micrometer